# An exploration of the professional identity of clinical academics using repertory grid technique

Elaine Burke[1]*, Karen Misstear[2], Martina Hennessy[1,2]

**1** School of Medicine, Trinity College Dublin, Dublin, Ireland, **2** Wellcome/Health Research Board Irish Clinical Academic Training Programme, Dublin, Ireland

* burkee11@tcd.ie

**Data Availability Statement:** All relevant data are contained within the paper and Supporting Information files.

**Funding:** The authors received no specific funding for this work.

## Abstract

### Background

Clinicians who divide their time between clinical work and research have contributed to some of the most fundamental breakthroughs in medicine in recent history, yet their role is not always well-understood or valued. Understanding the factors which contribute to career success for clinical academics is critical for supporting this workforce. Social Cognitive Career Theory (SCCT) provides a conceptual framework for career success, incorporating personal and environmental factors.

### Purpose

The aim of this study is to explore clinical academics' construal of successful clinical academic practice and to contribute to a holistic view of the professional identity of the clinical academic.

### Methodology

Using a constructivist technique, repertory grid, the authors interviewed ten clinical academics at different career stages in one-to-one structured interviews conducted virtually between November 2020 and April 2021. Data from the interviews were analysed qualitatively and quantitatively. Common themes were identified, analysed, and ranked according to importance with respect to successful clinical academic practice. Using SCCT as a framework, constructs were categorised as personal factors, organisational factors, competencies and person-environment fit. A differential analysis between established/trainee and female/male participants was carried out.

### Summary of results

One hundred and thirty-three constructs were elicited and categorised into 20 themes (constructs). There was consensus among participants that 6 were of high importance with respect to successful clinical academic practice, 8 of intermediate and 4 of low importance, with no consensus on 2 constructs. Personal factors of high importance include innovation

**Competing interests:** The authors have declared that no competing interests exist.

and integrity. Competencies including research and teaching skills are highly important, and ability to collaborate is also considered central to successful clinical academic practice. Female participants expressed greater concerns about the impact of familial responsibilities on career progression.

## Discussion and conclusions

This study highlights the importance of interactions between the person and environment, and characterises the important attributes of successful clinical academics including personal factors such as integrity and innovation.

## Introduction

Increases in clinical research funding over the past number of decades have not seen a parallel increase in cures and treatments for human disease, and the inability to translate basic scientific findings into treatments, combined with other inefficiencies, results in a wastage of up to 85% of research funding, estimated to cost $200 billion annually [1, 2]. Clinical academics, or clinician scientists combine clinical practice with research, are well-positioned to offer critical insights and identify pertinent research questions, helping to bridge the gap between research and patient care. They have contributed to some of the most fundamental breakthroughs in medicine in recent history [3], and research led by MDs is twice as likely to involve humans as PhD-led research [4]. However, despite the importance of the role, the clinical academic workforce is under threat, and there have been international calls to build and sustain it for the future [5–8].

The problems facing clinical academics are well-documented, and there is a growing body of literature providing strategies and approaches on supporting this crucial workforce [9, 10]. However, the literature primarily focusses on organisational factors, such as the availability of training programmes, and there is little information available on the professional identity of clinical academics, and the personal characteristics that support a successful career. Current definitions of the clinician scientist role tend to focus exclusively on research funding obtained, a superficial definition which does not acknowledge the many facets of this complex role [11]. Maintaining and supporting the clinician academic workforce requires a deeper understanding of the abilities and attributes of clinical academics.

Recruitment and promotion in clinical research can tend to emphasise research outputs and metrics such as the h-index or m-index, which are also associated with the ability to secure grant funding [12, 13]. In balancing their time between clinical work and research, clinical academics may struggle to meet the target metrics of full-time researchers. While many institutions are embracing the Declaration on Research Assessment (DORA) principles, moving away from using journal-based metrics to evaluate the quality of research and favouring a more descriptive portfolio [14], it is not yet known how this change will affect clinical academics who balance many responsibilities.

Professional identity in physicians has been defined as a set of characteristics, values and norms which become internalised over time; eventually the individual thinks, acts and feels like a physician [15]. Understanding professional identity and how it is formed requires an exploration of key attributes and determinants of success or failure [15, 16], although this only forms part of the picture. Competency frameworks have been developed which show a more rounded view of the work of clinical academics [17], however, there remains a gap in the

knowledge on how clinical academics develop a professional identity which integrates the dual disciplines of research and clinical practice, and what the critical determinants of success or failure are. Describing the personal and environmental factors that support a successful clinical academic career will contribute to the understanding of the professional identity of clinical academics, which may be a critical determinant of career stability; further, a greater understanding of the professional identity of clinical academics may inform selection criteria for recruitment and promotion and the development of education and training strategies to support the clinical academics of the future [16].

## Social cognitive career theory and career success

Considering the factors which help determine career success requires an understanding of what is meant by career success. There are many definitions of career success which have evolved over time from objective measures such as traditional hierarchical progression (climbing the ladder), to incorporate subjective components of how an individual views his/her own success in the context of personal needs and values. Therefore, interpretations of career success vary between individuals. Social Cognitive Career Theory (SCCT, Lent et al. 1994) combines internal and external factors (personal and environmental) which can help predict career success and provides a useful framework for conceptualising career development interventions [18, 19].

Social Cognitive Career Theory (SCCT, Lent et al. 1994), founded on the principles of Bandura's Social Cognitive Theory, explains how personal and environmental factors all interact to shape career development [18, 20]. There is extensive evidence to support SCCT as a framework for understanding career-related behaviour and outcomes and for measuring career success. It has also been shown to provide an effective theoretical foundation for exploring clinical academic and clinician educator career development and professional identity formation [21–23].

According to SCCT, career interest develops when learning experiences arise from person-environment interactions. Personal factors influence the nature and variety of learning experiences to which an individual is exposed. External factors such as supportive environmental conditions combine with personal factors to help shape career interest and determine success. Based on this model, four factors are proposed which can help predict career success: personal factors, organisational factors, learning experiences or competencies, and person-environment fit [18].

## Methods

### Research approach: Personal construct theory and repertory grid technique

Repertory grid technique (RGT) is an interview technique based on Personal Construct Theory (PCT). PCT is the idea that individuals construct the meaning of their own lives, and we are constantly creating, testing, and refining our personal theories, or construct systems, to make sense of the world around us and our place in it. Individuals construct their own version of reality using a hierarchical system in which both the constructs themselves and the hierarchy into which they are organised are personal: some constructs are more important, or central, others are less important, or peripheral [24, 25].

RGT is a structured interview procedure that allows an investigation into an individual's personal construct system and its hierarchical structure. Researchers are provided with insight into the unique and subjective experience of the interviewee, at the same time it is possible to numerically compare grids and draw universal conclusions. RGT has advantages over other approaches in that it combines both qualitative and quantitative methods to offer a robust

analysis, it eliminates interviewer bias, and it gathers data not only on constructs (in this case, factors required for success as a clinical academic), but the relative importance of each factor to the topic in question. It is the most widely used and researched constructivist technique to date, and numerous studies have confirmed its reliability by showing that constructs tend to remain stable over time [24].

## Participants

A recent qualitative study of perceptions of physician-scientists' success identified differences in the views of clinical academics depending on gender and career stage [26]. Therefore, to ensure views were representative of the group, purposive sampling was carried out and a balance between established and trainee clinical academics and male and female clinical academics was maintained (Table 1). Established clinical academics were supervisors and directors on the Irish Clinical Academic Training (ICAT programme), a combined clinical and academic training programme for higher specialist trainees. They have stated their identity as clinical academics on the ICAT website and have provided a CV for that purpose. Trainee clinical academics have successfully competed to enter a funded combined clinical academic training programme and are at Higher Specialist Trainee (HST) level. Thus, different career stages from higher specialist training to full professorship were represented.

A list of established and trainee clinical academics was drawn up with the help of a director and programme manager of the ICAT programme and potential participants were invited to interview. Two of the clinical academics we invited declined to participate; one because they felt they were too junior to be able to discuss the topic in detail, and the other because of time constraints. Participants represented 10 different specialties and 2 healthcare systems (the Irish Health Service Executive, HSE, and the UK's National Health Service, NHS).

## Interview technique

Interviews took place between November 2020 and April 2021. Informed written consent was obtained from all participants. Ten interviews were conducted virtually using downloadable computer software for video communications, ZOOM®, in line with government COVID regulations at the time which precluded face to face meetings, and audio from each interview was recorded. Interviewees chose the location of the interview and only the interviewer and interviewee were present. Interviews lasted between 49 minutes and 1 hour 22 minutes. Interviews were transcribed verbatim by the interviewer; transcripts were available to participants on request. Data was entered into GridSuite 5 for Windows (Institut für Personal Construct Psychology, Stuttgart, Germany [27]), a repertory grid software package which is freely available online, in real time, and the interviewee's grid was visible to them throughout the interview.

A pilot interview with an established clinical academic was initially carried out.

**Table 1. Participant demographics.**

|  | Established | Trainee |
|---|---|---|
| Female | 3 | 2 |
| Male | 2 | 3 |
| Total | 5 | 5 |

Approval for the study was granted by the Trinity College Dublin School of Medicine Research Ethics Committee in May 2020 (Ref. 20200201).

At the beginning of the interviews, participants were asked to think of six to eight clinical academics known to them. They were asked to consider of a range of effectiveness, i.e., to include some academics who they consider to be excellent, some they consider very good or average, and finally one or two who they would consider less effective than the others. These are known as the elements. The criteria for inclusion of elements were not strictly described because doing so would impose the interviewer's perceptions of success on the participants. Participants did not disclose the names of the elements to the interviewer, they assigned elements a letter A-H and referred to them only by their assigned letter. The interviewer therefore was not aware of the identities of the elements. This permitted a more open discussion and avoided concerns about identification of elements in a potentially negative way. A final element was included, the Ideal, an imaginary clinical academic who excels in all areas according to the participant's construal of excellence. The Ideal element provides additional contrast to help ensure important constructs are not missed.

Constructs were then identified using a procedure known as triadic elicitation. Once elements A-H and Ideal were entered into the software, the programme selected three at random. Participants were then asked the following question:

> *"With respect to successful clinical academic practice, please identify a way in which two of these elements are similar but different to the third."*

The response provided is referred to as the emergent pole; participants are then asked, *"What is the opposite of this*?*"*, with the response becoming the implicit pole. Much of the individual nuance of a construct is contained within the implicit pole, e.g., "good mentor" could be contrasted with "not interested in mentoring" or "unsupportive mentor", providing a slightly different meaning to the construct. Constructs were further explored to ensure clarity, contrast between the poles and a relationship to the concept of successful clinical academic practice.

Once a construct has been elicited, all elements are rated according to the construct on a 5-point scale with the emergent pole at the top end (1) and the implicit pole at the bottom (5), creating a numerical grid (Fig 1).

The entire process continues until no further constructs can be elicited. Once participants' constructs were exhausted, the interviewer provided five supplied constructs which participants could agree with, amend, or discard as they saw fit. This ensured their views on some commonly described attributes of clinical academics were included. The supplied constructs were based on clinical academic job descriptions, criteria used by major grant funders and findings from the pilot interview (Table 2).

Finally, participants were asked to rate all elements according to the construct "Overall, an effective clinical academic vs Overall, a less effective clinical academic". This enabled aggregate analysis of all 10 grids as described later.

In the example of a completed grid (Fig 1), elements A-Ideal are column headings and constructs are listed on the right-most and left-most rows, with the emergent pole on the left (1) and implicit pole on the right (5). Elements rated closest to the emergent pole are scored 1 whereas elements rated closest to the implicit pole are scored 5.

## Trustworthiness and credibility

All participants had an opportunity to review their grids and preliminary analysis at the end of the interview. No further constructs were elicited at this stage, and all agreed that the grids were an accurate representation of their construal of the topic of successful clinical academic practice.

Elements

| 1 (Emergent pole) | A | B | C | D | E | F | G | Ideal | 5 (Implicit pole) |
|---|---|---|---|---|---|---|---|---|---|
| Innovative | 1 | 1 | 4 | 3 | 4 | 1 | 5 | 1 | Unimaginative |
| Hardworking | 1 | 1 | 1 | 3 | 1 | 1 | 3 | 1 | Lazy |
| Honesty, integrity | 1 | 1 | 2 | 3 | 1 | 2 | 5 | 2 | Self-serving |
| High international impact | 1 | 1 | 2 | 1 | 4 | 1 | 5 | 1 | Lower international impact |
| Pure clinical research | 4 | 3 | 1 | 5 | 3 | 4 | 1 | 5 | Basic scientific research |
| Small research team | 5 | 5 | 1 | 5 | 1 | 4 | 1 | 5 | Large research team |
| Collaboration | 1 | 1 | 4 | 2 | 3 | 1 | 5 | 1 | Siloed, inward looking |
| Ability to acquire peer-reviewed funding | 1 | 2 | 3 | 1 | 4 | 1 | 5 | 1 | Reliance on soft funding |
| Common research infrastructure development | 1 | 1 | 5 | 1 | 2 | 3 | 3 | 1 | Own research infrastructure development |
| Altruism | 2 | 2 | 5 | 4 | 1 | 3 | 5 | 2 | Self-serving |
| Interest in teaching | 1 | 1 | 3 | 5 | 1 | 1 | 5 | 2 | Disinterested in teaching |
| Good mentor | 1 | 1 | 1 | 5 | 2 | 2 | 5 | 1 | Poor mentor |
| Focus on patient care | 3 | 3 | 2 | 5 | 4 | 3 | 2 | 3 | Research only focussed |
| Strong contribution to discipline | 1 | 1 | 1 | 3 | 2 | 1 | 5 | 1 | Weaker contribution to discipline |
| Commitment to deliver accurate information to the public | 1 | 3 | 2 | 3 | 4 | 1 | 5 | 1 | Providing less accurate information to the public |
| Overall, an effective clinical academic | 1 | 1 | 2 | 2 | 3 | 1 | 5 | 1 | Overall, a less effective clinical academic |

*(Left margin label: Constructs)*

**Fig 1. Example of a completed grid.**

After completion of 10 grids, constructs were reviewed, and it was agreed by both researchers involved in data analysis that data saturation had been reached and it was unlikely that further constructs would arise if additional interviews were conducted.

During the aggregate analysis described below, both researchers independently re-categorised all constructs according to the agreed category system. There was 90.23% agreement on where constructs lay within the revised category system, indicating a robust system [25].

## Researcher characteristics and reflexivity

All three authors are involved in combined clinical academic training programmes for junior doctors. The interviewer (EB) is not involved with the ICAT programme and did not have a

**Table 2. Supplied constructs.**

| Supplied Constructs |
|---|
| • Effective teacher/less effective teacher |
| • Clinically focussed/less clinically focussed |
| • Significant contribution to discipline/less significant contribution to discipline |
| • Strong collaborator/less likely to collaborate |
| • Strong public outreach/less focus on public outreach |

close working relationship with any of the interviewees. The technique of RGT was chosen as this essentially eliminates interviewer bias.

## Statistical analysis of individual grids

We calculated a % similarity score from the ratings for each construct by calculating the sum of differences between each possible pair of constructs within a grid and converting to a percentage score to facilitate comparison between grids (S1 File). Percent similarity scores for each element were calculated in a similar fashion. In calculating the % similarity score for constructs, ratings were reversed, and the highest % similarity score taken. This ensured no relationships were missed because constructs are bipolar and the allocation of the implicit and emergent pole (and hence the direction of the rating scale) is arbitrary [25].

## Aggregate analysis: Honey's content analysis

We conducted an aggregate analysis of all 10 grids using Honey's Content Analysis [25, 28], beginning with a core categorisation procedure and reliability check.

During the core categorisation procedure, similar constructs were grouped together under codes; the remaining items were then compared to existing codes and allocated accordingly. Similar codes were then grouped together under category headings or "Overarching constructs". The core categorisation was carried out independently by two researchers and initial codes compared. Following analysis and revision of the overarching constructs, a new category system was created, and both researchers re-categorised all constructs independently according to the agreed system. We conducted a final discussion and refinement of the category system until both researchers were satisfied with the categorisation.

We then allocated a H-I-L (High-Intermediate-Low importance) value to each construct according to the % similarity score between the construct in question and the supplied construct of "Overall, an effective clinical academic vs overall, a less effective clinical academic". The use of the H-I-L index (as opposed to using % similarity scores) reflects the fact that individual grids may differ in the range of their % similarity scores, so the % similarity scores are not directly comparable. We organised constructs into their Overarching construct, and H-I-L values for each were noted to see what the group has to say about a particular overarching construct. Overarching constructs with mostly H values are of particular importance to the group; mostly I or L values indicate constructs which are considered less central to effective clinical academic practice. Those where the H-I-L values are mixed indicate a high degree of ambivalence or diverse views towards a particular overarching construct.

## Differential analysis

Participants were divided evenly into two groups: established clinical academics and trainee clinical academics; they were also divided equally according to gender. The constructs of each group were analysed separately according to the aggregate analysis procedure described above to explore whether differences the construal of effective clinical academic practice exist between the groups.

## Results

### Aggregate analysis findings

Excluding the supplied summary construct of "Overall, an effective clinical academic vs overall, a less effective clinical academic", 133 constructs were elicited. These constructs were

grouped together into 57 codes and further organised into 20 overarching constructs (Table in S2 File).

There was a group consensus that 6 overarching constructs were of high importance, 8 were of intermediate importance and 4 were of low importance. There was no consensus regarding the importance of two overarching constructs, Clinical focus vs research focus and Working excessive hours vs good work-life balance, i.e., the H-I-L indices were mixed. Overarching constructs were sorted into four categories predictive of career success in keeping with the theoretical framework, SCCT: Personal Factors (PF), Organisational Factors (OF), Competencies (C) and Person-Environment Fit (PEF). Two constructs were not readily identifiable as belonging in a single category, incidentally these were the same two constructs for which there was no group consensus on their relative importance, i.e., there was a high degree of ambivalence. Table 3 below illustrates overarching constructs, their importance to the group, and the category of factors predictive of career success according to SCCT.

## Differential analysis findings

Established and trainee clinical academics provided similar numbers of constructs (68 and 65 respectively). Established clinical academics rated the constructs relating to innovation, commitment to public outreach, agreeableness, and democratic leadership style as being of higher importance compared to trainee clinical academics. Trainees put higher importance on the

**Table 3. Overarching constructs, importance, and category according to factor predictive of career success.**

| Overarching construct | Importance | Category |
|---|---|---|
| Research outputs have significant impact vs outputs have lower impact | High | C |
| Works to build a network of collaborators vs prefers to work alone | High | PEF |
| Established researcher, well-recognised in scientific community vs not well-recognised as a researcher | High | PEF |
| Innovative, embracing new ideas vs lacking imagination, closed-minded | High | PF |
| Honesty and integrity vs self-serving | High | PF |
| Excellent, inspiring teacher vs poor, boring teacher | High | C |
| Willing to help others, altruistic vs focussed on own goals | Intermediate | PEF |
| Agreeable, approachable vs antagonistic, intimidating | Intermediate | PEF |
| Access to resources vs less access to resources | Intermediate | OF |
| Committed to public outreach vs not committed to public outreach | Intermediate | C |
| Dedicated and hardworking vs lazy, inefficient | Intermediate | PF |
| Pure clinical research vs basic scientific research | Intermediate | C |
| Synergy between clinical and research work vs disconnect between clinical and research work | Intermediate | C |
| Good at communicating vs not good at communicating | Intermediate | C |
| Experienced researcher vs early career, less experienced | Low | PF |
| Democratic leader, fosters autonomy vs autocratic, micromanager | Low | PF |
| Greater demands on time outside work vs fewer demands on time outside work | Low | PF |
| Surgeon vs physician | Low | PF |
| Focus on clinical work vs focus on research | No consensus | None |
| Working excessive hours vs good work-life balance | No consensus | None |

PF = Personal Factor, OF = Organisational Factor, C = Competency, PEF = Person-environment Fit, NC = Not categorised.

constructs of "Willingness to help others vs. focussed on own goals" and "excessive working hours vs. good work-life balance" compared to established clinical academics. Both groups emphasised the importance of research outputs (S3 File, Table 1).

Female and male participants (who were divided evenly between the groups of established and trainee clinical academics) provided similar numbers of constructs (68 and 65, respectively). Male participants put greater importance than female on the constructs of agreeableness (although this arose more frequently among female participants), innovation and experience as a researcher. The construct of "Greater demands on time outside work vs fewer demands on time outside work", mostly reflecting family responsibilities, occurred exclusively among the female participants (S3 File, Table 2).

## Discussion

By exploring the personal construct systems of established and trainee clinical academics and their experiences of working with other clinical academics, we have examined how they construe success. We have interviewed clinical academics at different career stages across multiple institutions on the island of Ireland, representing two different health systems (the National Health System in the UK and the Health Service Executive in Ireland) and multiple specialties. Findings are considered using SCCT as a framework which takes both personal and environmental factors into account and contributing to a holistic view of the characteristics that define the identity of the successful clinical academic.

### Personal factors

Personal factors are important for career success because they form part of an individual's background which determines how the individual interacts with and influences learning experiences and the environments to which they are exposed. Personal factors include demographics, personal resources and personality traits such as conscientiousness [29]. Combined with other factors, e.g., organisational, personal factors determine career development and career performance [18].

Seven of the 20 overarching constructs are personal factors (Table 3). Of these, the constructs pertaining to innovation and honesty/integrity are of greatest importance.

Innovation, i.e., the ability to create and implement solutions to healthcare problems is seen as important for healthcare practitioners [30, 31], and is fundamental to successful clinical academic practice. While innovation in healthcare is one of the major consequences of a clinical academic workforce, and research-active institutions have better patient outcomes [32], the ability to innovate is rarely mentioned as a key characteristic for clinical academics. Innovation potential is seldom used as part of the selection criteria for clinical academic training pathways, even though tools to measure creativity and innovation potential exist [31].

Similar studies looking at characteristics of clinical academics and academic leaders have found variable results in relation to the importance of innovation. While Daouk-Öyry et al. list Innovation, Creativity and Dedication as key competencies for academic physicians under the heading of Research Skills [17], Innovation only comes 16th out of a list of 38 positive leadership values ranked by Deans and CEOs of Academic Medical Centres (AMCs) as being important to their leadership [33]. Elsewhere, in a survey of American College of Surgeons Society of Surgical Chairs on the importance of certain personal traits leading to a chair position, only 4 participants out of 52 placed Creativity in the top 5 out of a list of 26 traits (innovation was not included in the list) [34]. While these results are at odds with our finding that innovation is central to successful clinical academic practice, this may be because the latter two studies were focussed on the importance of traits with respect to being an effective leader in an academic centre as opposed to a

successful clinical academic; further research is needed to understand clinical academics' construal of innovation, and its overall importance to success as a clinical academic.

Work done on the Big Five personality dimensions in physicians found a negative correlation between some facets of Conscientiousness, a highly desirable trait among healthcare professionals, and Innovation, possibly because lower Conscientiousness is associated with greater flexibility [31]. At the same time, Conscientiousness is positively associated with success in undergraduate medical school and academic success in general [35, 36]. Given the importance of innovation for clinical academics, it may be useful to review selection processes to clinical academic training programmes which emphasise undergraduate academic performance without incorporating any specific measures of creativity and innovation. Notably, no participants in this study made any reference to excellence in undergraduate studies as an attribute of successful clinical academics.

Participants described their ideal clinical academic as someone with the highest levels of honesty and integrity. Research integrity is critical in clinical research, and instances of misconduct can be career-ending. The literature, however, presents a mixed picture of the importance of integrity to success as a clinical academic. Gotian et al., interviewing 21 physician-scientists at different career stages identified 23 subjective and objective characteristics that were associated with success as a clinical academic, but honesty and integrity were not among the characteristics identified [26]. In contrast, almost all the 25 clinical academics interviewed in another study developing a competency framework for clinical academics mentioned the importance of professional ethics and integrity [17]. Another study of leaders of academic medical centres found that integrity was considered the single most important trait (out of a list of 38 positive values) by 9 out of 18 academic leaders, and every participant listed it in their top four. Integrity was correlated positively with other traits such as respect and inspiration, however it correlated negatively with seizing opportunities and results [33]. Clinical academics are rewarded more for their results (research outputs) than for any other characteristic, so the relationship between integrity and outputs may merit further investigation.

## Competencies

According to SCCT, learning experiences are critical to developing self-efficacy (belief in one's ability to perform career-related tasks) and outcomes expectations (the outcomes an individual expects to arise from pursuing a particular career-related activity). Self-efficacy and outcomes expectations both inform career goals and choices and ultimately influence performance [18, 20]. Since individuals who engage in learning experiences are more likely to develop competencies which are related to career advancement, competencies and learning experiences can be considered under the same heading [18].

Six of the 20 overarching constructs can be considered to fall under the heading of competencies. Of these, two were of high importance: constructs pertaining to significant research outputs and excellence in teaching.

Contributing significantly to knowledge in one's discipline, influencing public policy and transforming patient care was central to participants' construal of successful clinical academic practice. This is not surprising as research metrics remain key to promotion and the awarding of grants in academic medicine [12, 13].

Teaching is considered one of the four core competencies for clinical academics [17]. Teaching is central to the work of medical schools [37], and clinical academics are often relied on to deliver teaching, but current structures do not reward teaching commitment to the same degree as research outputs. Emphasising research success over other attributes such as teaching excellence may contribute to faculty attrition [38].

## Person-environment fit

Learning occurs through interactions with the environment [20]. Person-environment fit (PEF) occurs when there is compatibility between individuals and their work environment; it has several components including person-job fit (PJF), person-organisation fit (POF) and person-group fit (PGF). Studies have found a strong connection between PEF and employees' attitudes to their jobs [39].

Four of the overarching constructs may fall under the heading of PEF. Of these, the two constructs relating to collaboration and recognition in the scientific community are of high importance, and the constructs relating to altruism and agreeableness are of intermediate importance.

It has been shown that cross-disciplinary teams have better outcomes and produce work of higher scientific impact [40], and clinical academic trainees rate opportunities to expand their collaborative network highly [41]. Other studies have shown collaboration and relationship building are key measures of success as a clinical academic [26, 33]. Willingness to help others and agreeableness are of intermediate importance to our participants, suggesting a need for balance between helping others with their goals and retaining focus on one's own goals.

The four constructs in the PEF category could be considered under the sub-category of person-group fit (PGF). PGF refers to compatibility between individuals and their workgroups. This is thought to be of particular importance to clinicians given the highly collaborative nature of medical practice and professionalism, and one study has shown that medical staff with higher PGF have both higher job satisfaction and higher professional efficacy [39]. PGF has also been shown to be positively associated with innovation [42]. Extrapolating from these findings, it is possible to suggest that clinical academics with high PGF have higher professional efficacy. However, this study did not directly measure PGF so further research would be required to investigate this signal.

## Comparison of established and trainee, female and male clinical academics

Trainees emphasised the importance of good work-life balance, whereas established clinical academics placed greater importance on innovation, agreeableness, and public outreach. Both groups placed an equal importance on research outputs. These findings are in keeping with a study from the US comparing perceptions of success among physician-scientists of different career stages: this study found that junior physician-scientists were more likely to emphasise research outputs, whereas more senior staff placed more emphasis on legacy [26]. The legacy of established clinical academics in this study may be that they have made a difference to patients and the public through innovation and are recognised and accepted by their peers.

The construct relating to demands on time outside work arose only for female participants. This overarching construct comprised two lower-order constructs relating to gender and childcare responsibilities. Gender was included in this overarching construct because participants felt female clinical academics are more likely to require time away from work for family and caring responsibilities and this in turn can affect career progression.

Both male and female clinical academics can face challenges brought about by balancing career with family responsibilities including childcare. However, there is evidence of significant differences in childcare responsibilities between male and female medical trainees, and female trainees are significantly more concerned about the impact of having a child on their career compared to their male counterparts [43]. These concerns are not without foundation: female physicians with children have lower rates of employment and lower career success when measured in terms of publications, grants, scholarships and research activities [44]; and the lack of females in senior clinical academic positions has been well-documented [45, 46].

This study confirms the finding that concerns about the impact of family responsibilities on career progression are greater for female clinical academics. The issue did not arise at all for male participants. This may be because they are less concerned about balancing their career with family responsibilities or they do not see it as being important with respect to successful clinical academic practice. It is also possible that there was a degree of reticence in discussing the issue with a female researcher. However, the issue of gender balance in academic medicine is one that affects everyone and tackling it will require the involvement of both male and female clinical academics.

## Limitations

Although both researchers involved in data analysis felt that data saturation was reached, the sample size was small and therefore findings should be interpreted with caution.

Some constructs which would be typical of repertory grid studies of other professions were absent (e.g., "Intelligent vs. dull"). A construct relating to holding additional research qualifications might also have been expected. Constructs which are common to all elements will not arise during triadic elicitation, a potential drawback to the interview technique. Caution should therefore be employed when considering the absence of certain constructs.

Participants were not geographically dispersed although they represented two different healthcare systems. Views from clinical academics in other sociocultural locations are therefore absent which creates a limitation to our understanding of the clinical academic in other locations.

Finally, there are some slight differences in terminology: during the elicitation of constructs, participants were asked to think of a range of effectiveness, from clinical academics they admire to those they would consider less effective. Effectiveness is not the same as success, however in such a competitive environment, ineffective clinical academics would be unlikely to succeed. Nonetheless, greater consistency with terminology during elicitation of elements may have produced a variation in the findings in relation to career success.

## Conclusion

Much of the literature on clinical academic careers focusses on organisational factors such as access to resources, and personal factors tend to be ignored. This study has highlighted the importance of interactions between the person and environment and characterised the important attributes of successful clinical academics which include personal factors as well as learning experiences/competencies and person-environment fit.

A picture emerges of a successful clinical academic: they are collaborative, excellent teachers and well-recognised in their scientific communities. They are also highly innovative, and their research outputs have significant impact in their field. They uphold the values of honesty and integrity. They are willing to help others and are approachable, while also maintaining professional boundaries and retaining focus on their own goals. They are dedicated, communicate well, and are committed to high quality public engagement. The bar is set high for clinical academics; few professions demand such a high degree of knowledge and skill combined with the personal, relational, and moral values described by the participants in the study. This workforce requires appropriate support for clinical academics in line with these high expectations and the provision of adequate resources including a clear career pathway and combined clinical academic training.

## Supporting information

**S1 File. Percent similarity score calculation.**
(DOCX)

**S2 File. Core categorisation procedure.**
(DOCX)

**S3 File. Differential analysis Tables 1 and 2.**
(DOCX)

## Acknowledgments

The authors wish to acknowledge the contribution of Dr Deirdre O'Shea, University of Limerick Ireland, to the conceptualization of this study.

## Author Contributions

**Conceptualization:** Elaine Burke, Martina Hennessy.

**Data curation:** Elaine Burke, Karen Misstear.

**Formal analysis:** Elaine Burke, Martina Hennessy.

**Investigation:** Elaine Burke, Martina Hennessy.

**Methodology:** Elaine Burke.

**Project administration:** Elaine Burke.

**Software:** Elaine Burke.

**Supervision:** Martina Hennessy.

**Validation:** Elaine Burke, Martina Hennessy.

**Writing – original draft:** Elaine Burke.

**Writing – review & editing:** Karen Misstear, Martina Hennessy.

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
