## [Decision Letter · Decision Letter 0]

5 Sep 2022

PONE-D-22-16880An exploration of the professional identity of clinical academics using repertory grid techniquePLOS ONE

Dear Dr. Burke,

Thank you for submitting your manuscript to PLOS ONE. After careful consideration, we feel that it has merit but does not fully meet PLOS ONE’s publication criteria as it currently stands. Therefore, we invite you to submit a revised version of the manuscript that addresses the points raised during the review process.

We look forward to receiving your revised manuscript.

Kind regards,

Andrea Bernardes, Ph.D.

Academic Editor

PLOS ONE

Journal Requirements:

" Unfunded study"

Additional Editor Comments:

Dear authors, thank you for sending the paper to Plos one. In general, the experience brought is quite interesting and points to important elements that should be considered for clinical academic practice.

There are, however, small adjustments to be made, as follows:

- The introduction needs to be more in-depth, emphasizing the problem; the importance of the study is unclear.

- Please bring more elements that give more clarity to the study rationale.

- More than 60% of the references used in the text were published more than 5 years ago. Please update.

Kindly, Andrea

Reviewers' comments:

Reviewer's Responses to Questions

**Comments to the Author**

1. Is the manuscript technically sound, and do the data support the conclusions?

Reviewer #1: Yes

Reviewer #2: Yes

2. Has the statistical analysis been performed appropriately and rigorously? 

Reviewer #1: Yes

Reviewer #2: Yes

3. Have the authors made all data underlying the findings in their manuscript fully available?

Reviewer #1: Yes

Reviewer #2: Yes

4. Is the manuscript presented in an intelligible fashion and written in standard English?

Reviewer #1: Yes

Reviewer #2: Yes

5. Review Comments to the Author

Reviewer #1: Overall, this is a clear, concise, and well-written manuscript. The introduction is relevant and theory based. Sufficient information about the previous study findings is presented for readers to follow the present study rationale and procedures. The methods are generally appropriate. Overall, the results are clear and compelling even the sample size was small and therefore findings should be interpreted with caution. The authors make a contribution to the research in this area of investigation. Overall, this is a high quality manuscript.

Reviewer #2: Dear authors, I would like to thank you for the opportunity to evaluate your work. My considerations aim to contribute to the improvement of the manuscript. This is a research that aimed to explore clinical academics’ construal of successful clinical academic practice and to contribute to a holistic view of the professional identity of the clinical academic. For this, the grid repertory was used and a quantitative and qualitative analysis was proposed, anchored in the SCCT as framework. In my view, the introduction presents the necessary elements for the reader to understand the relevance of the study, as well as the relevance of evaluating the construction of physician-investigators's career and its unfolding in the field of teaching. Considering that the term professional identity is a rather broad concept and is structured differently depending on the theoretical foundation on which it is based, I think that the text would benefit from a more precise allusion to the concept, from the point of view of the theory that the authors adopt. The text implies that the professional identity of physician-investigators is imbued with personal characteristics and values, developed in interaction with organizational aspects, but the concept is not explicitly outlined. In this direction, it is necessary to consider that the professional identity is more than the sum of the characteristics of success and this could be better explored. The method is well written, and explains the measures to guarantee scientific rigor. With regard to the qualitative aspect of data collection and analysis,I suggest that some points be clarified according to the COREq guideline: Were only the interviewer and the interviewee present at the time of the virtual interview? Which virtual platform was used for the interviews? What precautions were established to ensure a safe environment for verbalizing aspects related to the career? Were there refusals to participate in the study? What were the reasons? Is there any characterization of the participants, in addition to gender, that deserves to be mentioned? How long did the interviews last? The results are presented appropriately and the discussion explores the main aspects. I consider it important that the authors point out that the elements of success versus insucess mentioned refer to a given well-defined socio-cultural location, this will naturally differ in other sociocultural realities of medical education (for example, developing countries). A note in this sense, in the discussion, would be very relevant, since the identified constructs concern a given historical and cultural reality.

6. PLOS authors have the option to publish the peer review history of their article (what does this mean?). If published, this will include your full peer review and any attached files.

Reviewer #1: No

Reviewer #2: No

---

## [Author Response · Author response to Decision Letter 0]

20 Oct 2022

Many thanks to Dr Bernardes and reviewers for their helpful feedback and comments, we greatly appreciate the time you have spent reviewing and considering this paper. We have addressed all the comments in the response letter and made amendments as recommended. 

I am including my response again below but for clarity it would be easier to read it in the response letter.

We look forward to receiving your response with regard to this revised manuscript.

Please ensure that your manuscript meets PLOS ONE's style requirements, including those for file naming 

I have reviewed the style requirements and amended the manuscript accordingly 

Page 1, line 8: corresponding author’s initials added after email address

Page 8, line 151: table caption moved above table

Page 10, line 195: figure caption added to end of paragraph

Page 10, line 202: table caption moved above table

Page 14, line 280-281: table caption moved above table

" Unfunded study"

 The authors received no specific funding for this work The amended statement is included in the cover letter

In your Data Availability statement, you have not specified where the minimal data set underlying the results described in your manuscript can be found. PLOS defines a study's minimal data set as the underlying data used to reach the conclusions drawn in the manuscript and any additional data required to replicate the reported study findings in their entirety. All PLOS journals require that the minimal data set be made fully available. For more information about our data policy, please see http://journals.plos.org/plosone/s/data-availability.

The minimal data set has been included in the supplementary information. This indicates the core categorisation procedure: readers will be able to see the codes that formed the overarching constructs and the frequency with which they arose. 

Page 26, line 599: Supporting information

S2 File: Core categorisation procedure

This file is cited on Pg 13, line 267

S2 file submitted with manuscript

Please include captions for your Supporting Information files at the end of your manuscript, and update any in-text citations to match accordingly. Please see our Supporting Information guidelines for more information: http://journals.plos.org/plosone/s/supporting-information.

Caption and in text citations added as described above 

I have reviewed the reference list and can confirm that it is complete and correct. Links to all papers have been checked and as of 12th October 2022, none have been retracted. 

The introduction needs to be more in-depth, emphasizing the problem; the importance of the study is unclear. 

I have rewritten the opening paragraph to emphasise the problem and the importance of the study

 Pg 4, Lines 47-57

Increases in clinical research funding over the past number of decades have not seen a parallel increase in cures and treatments for human disease, and the inability to translate basic scientific findings into treatments, combined with other inefficiencies, results in a wastage of up to 85% of research funding, estimated to cost $200 billion annually (1, 2). Clinical academics, or clinician scientists combine clinical practice with research, are well-positioned to offer critical insights and identify pertinent research questions, helping to bridge the gap between research and patient care. They have contributed to some of the most fundamental breakthroughs in medicine in recent history (3) and research led by MDs is twice as likely to involve humans as PhD-led research (4). However, despite the importance of the role, the clinical academic workforce is under threat, and there have been international calls to build and sustain it for the future (5-8). 

Please bring more elements that give more clarity to the study rationale.

The rationale of the study is further explained in the second and fourth paragraphs of the introduction 

Pg 4, lines 62-66

Current definitions of the clinician scientist role tend to focus exclusively on research funding obtained, a superficial definition which does not acknowledge the many facets of this complex role (11). Maintaining and supporting the clinician academic workforce requires a deeper understanding of the abilities and attributes of clinical academics.

Pg 5, lines 83-88

Describing the personal and environmental factors that support a successful clinical academic career will contribute to the understanding of the professional identity of clinical academics, which may be a critical determinant of career stability ; further, a greater understanding of the professional identity of clinical academics may inform selection criteria for recruitment and promotion and the development of education and training strategies to support the clinical academics of the future (16).

More than 60% of the references used in the text were published more than 5 years ago. Please update. 

I have repeated a literature review and included recent studies of relevance, e.g.:

Eshel et al, 2022; Bensken et al, 2019; Lent et al, 2019; Byram et al, 2022 

Considering that the term professional identity is a rather broad concept and is structured differently depending on the theoretical foundation on which it is based, I think that the text would benefit from a more precise allusion to the concept, from the point of view of the theory that the authors adopt. The text implies that the professional identity of physician-investigators is imbued with personal characteristics and values, developed in interaction with organizational aspects, but the concept is not explicitly outlined. In this direction, it is necessary to consider that the professional identity is more than the sum of the characteristics of success and this could be better explored. 

This paper explores the professional identity of the clinical academic which is made up of characteristics, values, and norms which are internalised over time. I have clarified what I mean by the term “professional identity”, as you say it is a broad term and can have different meanings depending on the context. Exploring the characteristics of a successful clinical academic can contribute to the understanding of the professional identity of clinical academics, but only forms part of the picture. I have expanded one paragraph of the introduction, and used phrases such as “contribute to the understanding” to emphasise that characteristics alone do not form a complete understanding of professional identity.

Social Cognitive Career Theory has recently been used to explore professional identity formation in clinician educators (Byram et al 2022), and a reference to this study has been added. Page5, lines 745-88

Professional identity in physicians has been defined as a set of characteristics, values and norms which become internalised over time; eventually the individual thinks, acts and feels like a physician (15). Understanding professional identity and how it is formed requires an exploration of key attributes and determinants of success or failure (15, 16), although this only forms part of the picture. Competency frameworks have been developed which show a more rounded view of the work of clinical academics (17), however, there remains a gap in the knowledge on how clinical academics develop a professional identity which integrates the dual disciplines of research and clinical practice, and what the critical determinants of success or failure are (16). Describing the personal and environmental factors that support a successful clinical academic career will contribute to the understanding of the professional identity of clinical academics, which may be a critical determinant of career stability ; further, a greater understanding of the professional identity of clinical academics may inform selection criteria for recruitment and promotion and the development of education and training strategies to support the clinical academics of the future (16).

Page 6, line 102-105

It has also been shown to provide an effective theoretical foundation for exploring clinical academic and clinician educator career development and professional identity formation (21-23). 

With regard to the qualitative aspect of data collection and analysis, I suggest that some points be clarified according to the COREq guideline: 1. Were only the interviewer and the interviewee present at the time of the virtual interview? 

2. Which virtual platform was used for the interviews? 

3. What precautions were established to ensure a safe environment for verbalizing aspects related to the career? 

4. Were there refusals to participate in the study? What were the reasons? 

5. Is there any characterization of the participants, in addition to gender, that deserves to be mentioned? 

6. How long did the interviews last 

I have clarified the points raised in the adjacent column.

In response to the fifth point raised, have now included the differential analyses we carried out between established and trainee clinicians, and female and male clinicians. This was initially omitted in the interest of brevity, however there were some interesting findings, so we have now included a summary of this analysis. Information has been added to the methods, results and discussion sections. The tables pertaining to this analysis are quite large so I chose to include them separately in a supplemental file. 

1. Page 8, lines 159-160

only the interviewer and interviewee were present 

2. Page 8, lines 156-157

Ten interviews were conducted virtually using downloadable computer software for video communications, ZOOM®,

3. Page 8, lines 155-156

Informed written consent was obtained from all participants. 

Page 9, lines 173-177

Participants did not disclose the names of the elements to the interviewer, they assigned elements a letter A-H and referred to them only by their assigned letter. The interviewer therefore was not aware of the identities of the elements. This permitted a more open discussion and avoided concerns about identification of elements in a potentially negative way. 

4. Page 8, lines 147-149

Two of the clinical academics we invited declined to participate; one because they felt they were too junior to be able to discuss the topic in detail, and the other because of time constraints. 

5. Multiple additions:

Page 2, line 30-31

A differential analysis between established/trainee and female/male participants was carried out. 

Page 3, line 38-40

Female participants had greater concerns about the impact of familial responsibilities on career progression.

Page 13, lines 258-262 (paragraph under heading “Differential analysis”)

Page 15, lines 285-298 (paragraph under heading “Differential analysis findings”)

Page 20, lines 411-439 (paragraph under heading “Comparison of established and trainee, female and male clinical academics”)

Tables to illustrate the findings from the differential analysis are included in a supplemental file, S3 File

6. Page 8, line 160

Interviews lasted between 49 minutes and 1 hour 22 minutes.

I consider it important that the authors point out that the elements of success versus insucess mentioned refer to a given well-defined socio-cultural location, this will naturally differ in other sociocultural realities of medical education (for example, developing countries). A note in this sense, in the discussion, would be very relevant, since the identified constructs concern a given historical and cultural reality. 

We agree this is an important point, and is now included in the discussion under the Limitations heading. 

Page 21, Lines 449-452: Participants were not geographically dispersed although they represented two different healthcare systems. Views from clinical academics in other sociocultural locations are therefore absent which creates a limitation to our understanding of the clinical academic in other locations.

---

## [Editor Report · Decision Letter 1]

26 Oct 2022

An exploration of the professional identity of clinical academics using repertory grid technique

PONE-D-22-16880R1

Dear Dr. Burke,

We’re pleased to inform you that your manuscript has been judged scientifically suitable for publication and will be formally accepted for publication once it meets all outstanding technical requirements.

Kind regards,

Andrea Bernardes, Ph.D.

Academic Editor

PLOS ONE

Additional Editor Comments (optional):

Dear authors,

The paper has been greatly improved and is therefore accepted for publication. Congratulations on the study!

Kindly, Andrea Bernardes.
---

## [Editor Report · Acceptance letter]

8 Nov 2022

PONE-D-22-16880R1 

An exploration of the professional identity of clinical academics using repertory grid technique 

Dear Dr. Burke:

I'm pleased to inform you that your manuscript has been deemed suitable for publication in PLOS ONE. Congratulations! Your manuscript is now with our production department. 

Kind regards, 

on behalf of

Dr. Andrea Bernardes 

Academic Editor

PLOS ONE